# Comparison of Cardiovascular Risk Factors among Somalis Living in Norway and Somaliland

**DOI:** 10.3390/ijerph16132353

**Published:** 2019-07-03

**Authors:** Soheir H. Ahmed, Niki Marjerrison, Marte Karoline Råberg Kjøllesdal, Hein Stigum, Aung Soe Htet, Espen Bjertness, Haakon E. Meyer, Ahmed A. Madar

**Affiliations:** 1Department of Community Medicine and Global Health, Institute of Health and Society, University of Oslo, 0318 Oslo, Norway; 2College of Medicine & Health Science, University of Hargeisa, 002563 Hargeisa, Somaliland; 3Health Services Research, Norwegian Institute of Public Health, 0213 Oslo, Norway; 4International Relations Division, Ministry of Health and Sports, Nay Pyi, Taw 15011, Myanmar; 5Division of Mental and Physical Health, Norwegian Institute of Public Health, 0473 Oslo, Norway

**Keywords:** Somalis, comparison, cardiovascular disease risk factors, Framingham risk score

## Abstract

Objective: We aimed to assess and compare cardiovascular disease (CVD) risk factors and predict the future risk of CVD among Somalis living in Norway and Somaliland. Method: We included participants (20–69 years) from two cross-sectional studies among Somalis living in Oslo (*n* = 212) and Hargeisa (*n* = 1098). Demographic data, history of CVD, smoking, alcohol consumption, anthropometric measures, blood pressure, fasting serum glucose, and lipid profiles were collected. The predicted 10-year risk of CVD was calculated using Framingham risk score models. Results: In women, systolic and diastolic blood pressure were significantly higher in Hargeisa compared to Oslo (*p* < 0.001), whereas no significant differences were seen in men. The ratio of total cholesterol to high-density lipoprotein (HDL) cholesterol was significantly higher in Hargeisa compared to Oslo among both men (4.4 versus 3.9, *p* = 0.001) and women (4.1 versus 3.3, *p* < 0.001). Compared to women, men had higher Framingham risk scores, but there were no significant differences in Framingham risk scores between Somalis in Oslo and Hargeisa. Conclusion: In spite of the high body mass index (BMI) in Oslo, most CVD risk factors were higher among Somali women living in Hargeisa compared to those in Oslo, with similar patterns suggested in men. However, the predicted CVD risks based on Framingham models were not different between the locations.

## 1. Introduction

Cardiovascular diseases (CVDs) remain the leading cause of death worldwide and pose a steadily increasing burden throughout low- and middle-income countries, including throughout sub-Saharan Africa [1,2,3]. The majority of CVD is caused by the presence and complex interaction of various modifiable risk factors, including hypertension, diabetes, dyslipidaemia, overweight and obesity, tobacco smoking, unhealthy diet, and harmful use of alcohol [4].

In sub-Saharan Africa, economic development has been linked to globalization and urbanization, which have been identified as root causes of a nutrition transition and the increasing prevalence of CVDs and their risk factors [5]. Somaliland is a region in the Horn of Africa with a population of approximately 4.5 million people that is currently undergoing such urbanization and epidemiological transitions [3,6]. However, we are not so far aware of any study reporting on the prevalence of CVD and its associated risk factors in Somaliland [7].

The ongoing conflict in the Horn of Africa has also led to a large diaspora of Somalis around the world [8]. Many Somalis have migrated to Norway, where they make up one of the largest immigrant groups from a non-Western country [9]. Previous studies on immigrants in Norway have had heterogeneous results regarding CVD risk factors [10,11,12]. However, despite the concentration of Somalis in Norway and the suggestions of increased CVD risk among ethnic minority groups, little research has been conducted on CVD risk factors among Somalis living in Norway [13].

Models such as the Framingham risk score have been developed in an effort to use data that can be easily collected to assess the combined effect of various risk factors on the risk of experiencing future CVD events [14,15]. However, the Framingham risk score has been criticized for both overestimating [16,17] and underestimating the absolute risk in different socioeconomic groups [18] and populations. Nonetheless, the Framingham risk score has been validated for use in various populations, including those of African origin living in the United States [14,19], and it also remains a valuable tool for disease projection estimates where there are only limited cross-sectional data available [20].

Unfortunately, the scarce data specific to Somalis limit the means and incentive for effective and increasingly necessary CVD prevention, intervention, and treatment [1,21]. We have previously reported high body mass index (BMI) and high prevalence of overweight and obesity among Somali men and women in Oslo compared to their counterparts in Hargeisa [22]. The aim of the current analysis is to assess and compare CVD risk factors and predict future CVD risk among Somalis living in Hargeisa, Somaliland and in Oslo, Norway.

## 2. Materials and Methods

In the present comparative study, we included data from two cross-sectional studies among Somalis living in Norway (Oslo study) and Somaliland (Hargeisa study), conducted between December 2015 and October 2016 and between March and September 2016, respectively. In both studies, participants were excluded if they were confirmed pregnant or if they were suffering from kidney or liver failure or cancer.

### 2.1. Oslo Study

The Oslo study was conducted in the Sagene district, which was selected because it has one of the highest populations and concentrations of people of Somali origin living there—approximately 1200 persons in total [9]. We established cooperation with user partners such as Somali organizations, a healthy life center, a volunteer center, the district medical officer, the community development center, and the local Somali radio station to share information about the study. We attempted to contact every adult person of Somali origin living in the district, and those available were invited to participate in the study. A total of 271 eligible men and women aged 20–69 years were invited, resulting in a final sample of 111 women and 110 men. Fifty persons who were contacted either did not want to participate or did not come for the appointment.

### 2.2. Somaliland Study

Due to the lack of data on the prevalence of risk factors in the population under study, the sample size was calculated using the diabetes prevalence of 4% [23]. There is no population registry in Somaliland, and the only available registry is the number of households. Each household has a unique number. Hargeisa city comprises five major districts which are each further subdivided into four main subdistricts. The sample design was two-stage cluster sampling, and the twenty subdistricts were targeted as the primary sampling units (PSUs). Ten subdistricts were randomly selected from the twenty PSUs (first stage), and a total of 1100 households were randomly selected from them (second stage) based on the probability proportionate to size (PPS) in each subdistrict.

In each selected household, all eligible persons aged 20–69 years living in the house were listed on a Kish household coversheet [24]. Men and women were listed in order of decreasing age and given a rank number. If the selected person rejected participation or was not present at home after two attempts, the next person on the Kish list was selected until there was one person from each household participating in the study. If there was nobody at home on the day of the study, a notification card was left at the door, and we returned the next day until we had one participant from each house. Data collection continued until there were 1100 participants, resulting in a final sample of 955 women and 145 men.

### 2.3. Data Collection

The two studies followed similar data collection methods and used the same tools. The Hargeisa study followed the World Health Organization (WHO) STEPwise approach [25]. Interviewer-administered questionnaires were used to collect information on years of education, occupation, marital status, personal medical history, medication use, smoking habits, and alcohol consumption. Participants were asked if they consume alcohol such as beer, wine, spirits, or other alcohol types (yes/no). They were also asked about the type of oil used for food preparation (such as cooking, frying, and baking).

In both studies, weight and height were measured with participants standing without shoes and wearing light clothing. Body weight (kilograms) was recorded to the nearest 0.1 kg and measured using an Omron medical scale that was checked every day with a known weight. Height (centimeters) was recorded to the nearest 0.5 cm and measured with a manual height-measuring instrument (SECA stadiometer) with participants standing upright with the head in the Frankfort plane. Body mass index (BMI) was calculated as weight in kilograms divided by the square of the height in meters (kg/m^2^).

Blood pressure (mmHg) was measured three times at one minute intervals using a validated automatic device (Omron HBP 1300) [26] with appropriate cuffs in a sitting position after at least 5 min of rest. The mean of the second and third readings was used for analysis.

### 2.4. Blood Sampling and Laboratory Assays

Fasting venous blood samples were collected from participants to determine the concentrations of serum glucose and lipids (fasting serum glucose (FSG), total cholesterol (TC), high-density lipoprotein (HDL) cholesterol, low-density lipoprotein (LDL) cholesterol, and triglycerides (TG)).

In Hargeisa, samples were collected at Hargeisa Group Hospital. Blood samples were collected in serum separator gel tubes and centrifuged after 30 min. Serum and plasma were then separated and frozen in aliquots at −30 °C the same day until being shipped to Oslo (Norway) where they were analyzed as one batch at the Fürst Medical Laboratory (http://www.furst.no), which has been accredited by the Norwegian Accreditation according to the standard NS-EN ISO 15189 TEST 209. The inter-assay coefficients of variation were 1.3% (TC), 1.6% (LDL cholesterol), 1.8% (HDL cholesterol), and 3.8% (TG). Total cholesterol, LDL cholesterol, HDL cholesterol, and TG were measured using an enzymatic method (ADVIA 2400 Siemens, produced by Jeol, Japan). In Oslo, blood samples were collected at the downtown office, Fürst Medical Laboratory, and were analyzed there within the same day, following the same procedure.

### 2.5. Definitions of Risk Factor Variables

Age was defined as completed years of age. Current tobacco smoking and alcohol consumption were ascertained by self-report as yes/no. Overweight was defined by a BMI of ≥25 kg/m^2^ [27]. Hypertension was defined as systolic blood pressure (SBP) of ≥140 mmHg and/or diastolic blood pressure (DBP) of ≥90 mmHg and/or being on blood-pressure-lowering medication [7]. Diabetes was defined as fasting serum glucose (FSG) of ≥7.0 mmol/L and/or being on medication [4,28]. High cholesterol was defined as a ratio of total cholesterol (TC) to high-density lipoprotein (HDL) of ≥5 and/or being on cholesterol-lowering medication [29].

### 2.6. Statistical Analysis

Continuous variables were expressed as means (SD), and percentages were used to express categorical variables. Significance was tested for continuous variables using independent *t*-tests and for categorical variables by chi-square test. The age-adjusted mean and prevalence were calculated by group and gender using predictive margins based on the linear and logistic regression models. To control confounding by age in the predicted means and prevalence, we fixed age at age equal to 40 years (which was closest to the mean age). Additionally, we tested for interaction between age and gender and between age and location (Hargeisa versus Oslo), assessing the marginal effect on lipid and BMI Framingham risk score models by age (Figure 1 and Figure 2).

Data were analyzed using Stata statistical software, version 14.0 (StataCorp. College Station, TX, USA). The results were considered statistically significant with *p* < 0.05.

### 2.7. Estimating the Framingham Risk Score

The predicted 10-year risk for an incident cardiovascular event was estimated using the Framingham 10-year risk score models, as published by D’Agostino et al. in 2008 [15]. The risk score includes age, smoking status, systolic blood pressure, diabetes, total cholesterol, and HDL cholesterol [30], alternatively with BMI replacing total cholesterol and HDL cholesterol [15,30]. We used both the lipid-based and BMI-based models for the gender-specific algorithm. Individuals with prior CVD were excluded from these analyses, and only those with complete blood samples were included in the lipid-based model and analyses.

For the lipid-based equation, we included a total of 123 men (73 from Oslo and 50 from Hargeisa) and 552 women (82 from Oslo and 470 from Hargeisa). In the BMI-based equation, a total of 235 men (101 from Oslo and 134 from Hargeisa) and 963 women (103 from Oslo and 860 from Hargeisa) were included (Figure 3). The age-adjusted mean risk scores and percentages (95% CI) of those with a predicted 10-year risk of ≥10% are presented. In additional analyses, Framingham risk scores were stratified by level of education.

Both studies were approved by the Regional Committee for Medical and Health Research Ethics Norway (study codes: 2015/1552 and 2015/2448 REK South-East). The Ministry of Health in Somaliland additionally approved the Somaliland study. In both studies, written informed consent was obtained from all participants.

## 3. Results

A total of 1310 participants were included in the analysis. Women from Hargeisa composed the largest subgroup with 953 participants, while 106 women participated from Oslo. Among men, there were 145 participants from Hargeisa and 106 from Oslo.

The percentage of participants with a high level of education, defined as having completed secondary school or more, was highest among men living in Oslo (89%) and lowest among women living in Hargeisa (14%) (Table 1). Unemployment, in turn, was lowest among men living in Oslo (19%) and highest among women living in Hargeisa (88%). Homemakers identifying as unemployed likely inflated this figure among women in Hargeisa.

The mean SBP and DBP were significantly higher in women in Hargeisa than in women in Oslo (*p* < 0.001) with the same pattern suggested in men, although differences were not significant (Table 1). The mean TC/HDL cholesterol ratio was significantly higher among women and men in Hargeisa when compared to men and women in Oslo (Table 2). There was no significant difference in the mean levels of FSG or TC between the two locations, but triglycerides were higher in Hargeisa (Table 2).

The prevalence of hypertension was significantly higher among women from Hargeisa (35%) compared to among women from Oslo (17%, *p* < 0.001). In men from both Hargeisa and Oslo, the prevalence of hypertension was over 30% (Table 3). The prevalence of high cholesterol was significantly higher in Hargeisa than in Oslo in both genders (Table 3). The prevalence of diabetes among men tended to be higher in Hargeisa compared to Oslo, but the differences were not significant (Table 3).

Overweight/obesity prevalence was significantly higher among men in Oslo than in Hargeisa and was similarly higher among women in Oslo than in Hargeisa (Table 3). No women or men in Hargeisa reported any alcohol consumption, although 9% of men in Oslo did. No women in either location reported currently smoking, while the smoking prevalence was 27% among men in Hargeisa and 20% among men in Oslo (Table 3).

In an additional analysis, we found that 97.8% of Somalis in Hargeisa used palm oil for food preparation such as cooking, frying, and baking. In contrast, 76.6% of Somalis in Oslo used either olive, rapeseed, or sunflower oil in food preparation.

We also compared BMI, SBP, and DBP in those who provided a blood sample and those who did not. However, BMI, SBP, and DBP did not differ significantly between these two groups. The only exception was that BMI was significantly different in men in Oslo (data not shown).

### Predicted Framingham Risk Score

The mean predicted Framingham risk scores increased substantially with age (Figure 1 and Figure 2). The mean predicted risk was not significantly different in men from Hargeisa and Oslo (*p* = 0.98 for the lipid-based score and *p* = 0.70 for the BMI-based score). Similar results were found among women (*p* = 0.61 for the lipid-based score and *p* = 0.83 for the BMI-based score). However, compared to women, men had significantly higher risk scores (*p* < 0.001 the lipid-based score and *p* < 0.001 for the BMI-based score). The estimated proportions with a predicted CVD risk of ≥10% were not significantly different between the two locations, although it tended to be higher in men from Hargeisa (Table 4). Additionally, there was no significant relationship between the lipid-and BMI-based Framingham risk scores and educational level in men and women (Table 5). We ran the same analysis specified by location, which was not significant (data not shown).

## 4. Discussion

In spite of a higher mean BMI and a higher prevalence of overweight and/or obesity among participants in Oslo, there was no apparent difference in the predicted Framingham risk scores in our study populations. However, blood pressure was significantly higher among women in Hargeisa compared to women in Oslo, and in both genders, blood lipids tended to be more favorable in Oslo than in Hargeisa. To our knowledge, this is the first article to assess CVD risk factors and to estimate the predicted risk of CVD among Somalis living in Norway and in Somaliland.

Framingham risk scoring predicts the absolute risk of future CVD and is used to inform the clinical management of asymptomatic individuals. It also provides valuable insight into the cumulative effect of the included risk factors on future CVD occurrence in our study populations. Based on our results, men had a higher Framingham risk scores than women. This is in line with previous research and has been reported in other populations of African origin across the socio-economic spectrum and among immigrant groups in Norway [11,20]. The similar Framingham risk levels in Oslo and Hargeisa suggest that the sums of negative and positive factors are rather similar in the two locations.

There were differences in cardiovascular risk factors between the two locations and genders. Blood lipids were more favorable in Oslo than in Hargeisa. Men and women in Hargeisa had lower HDL cholesterol and a higher ratio of total cholesterol to HDL cholesterol when compared to men and women in Oslo. Mean triglyceride levels were also higher in Hargeisa than in Oslo, whereas the mean TC levels were not different between Oslo and Hargeisa. This is in accordance with previous results from a comparative study on Sri Lankans in Norway and Sri Lanka [31], where Sri Lankans in Oslo, despite their higher BMI, had a more favorable lipid profile when compared with their counterparts in Sri Lanka, possibly due to healthier nutritional practices and oil used after migration [32].

In our study, we found that Somalis in Oslo used healthier oil in food preparation than Somalis in Hargeisa, indicating that the type of fat used may play a role in the more favorable lipid profile in Oslo, as the type of fat (saturated versus unsaturated) affects blood lipid concentration and CVD risk [33]. Although we did not collect information on family income in any of the settings, the expectation is that those living in Oslo might have a higher income that can improve their access to a healthy diet and to foods with better nutritional value than those living in Hargeisa. Moreover, blood pressure was higher in women living in Hargeisa compared to Oslo, whereas similar differences were not evident in men. Comparing our findings with results from the Oslo immigrant study, the systolic blood pressure in Somali men in Oslo was rather similar to those of the other immigrant men. However, the Somali women in our study were in the lower range of systolic blood pressure when compared to women from Pakistan, Sri Lanka, and Turkey [12].

Smoking was non-existent in women, and in men, the prevalence estimate was higher in Hargeisa compared to Oslo, although this was not statistically significant. This contributes to the lower Framingham scores in women than in men as smoking is an important variable in the Framingham risk model as well as in lived cardiovascular health, and it is evidently a dire public health threat affecting the men in this population [34].

Despite the higher BMI among the Oslo participants, men and women from Hargeisa in this study tended to have a higher prevalence of diabetes, although the differences were not significant. In general, other studies have reported a strong relationship between BMI and increased risk of metabolic diseases such as diabetes mellitus [35,36]. While we do not fully understand this pattern, an explanation might be that there are additional environmental factors which increase the risk of developing metabolic conditions in Hargeisa, such as different nutrition opportunities and different access to health care.

No relationship was found between the mean scores of Framingham risk and educational level among the genders, which is contrary to the results of other studies [37,38]. Further research on the relationship between CVD and social determinants of health among Somalis in Hargeisa and Oslo, as part of the diaspora from Somaliland, would provide additional beneficial insight into this phenomenon.

### Strengths and Limitations

This study has several strengths. The study contributes to the limited body of research on cardiovascular risk factors among Somalis who live in the Horn of Africa and Somalis who live as part of the diaspora in Norway. Both studies used a similar design, the same standardized questionnaires, protocols, equipment, and the same laboratory for blood sample analysis in Oslo, Norway, which facilitated comparison between the two studies. As this paper included major CVD risk factors and predicted the combined effect of risk factors using the Framingham risk scores, these results might be of interest both to public health and in clinical settings.

However, we recognize a number of limitations when interpreting these results. In the Hargeisa study, there is under-representation and possible selection bias among the men. In our study, we used the Kish grid, as the Kish grid addresses the selection of gender and age in a sample, but there is discussion as to whether the Kish grid can provide a representative sample for gender [39]. During the study, more women than men were in households. Moreover, 50 eligible men refused or were not available after several attempts at contact. If the selected person was not at home, we left a note that our team would come back the next day, and if they were not present after a second attempt, we selected the next eligible person from the Kish grid. According to Somali culture, women are often at home during the daytime while men are away working or socializing with other men. Therefore, there might have been a selection bias among men who were home and included in the study, as they may have been home and willing to participate for special reasons such as poor health or lack of work. Again, results pertaining to the low number of men in this study should be treated cautiously.

There is also a possibility that the Oslo study suffered some selection bias. We attempted to recruit all eligible individuals, and those who did participate may have had personal motivations for doing so. On the other hand, the proportion of those who participated is unclear as the living addresses of immigrants are often not up to date in the Norwegian Population Registry. Nevertheless, previous analysis of self-selection bias in studies of immigrants in Oslo suggests that it does not influence prevalence estimates to any great degree [40], although that study did not focus on Somalis.

The low number of participants who participated in the blood sample analysis is a possible limitation of this study. Although this was offered to all participants, ultimately, 72% of those in Oslo and 54% of those in Hargeisa took part. This might have been due to negative perceptions towards blood sampling in Somali culture or the location of the blood collection service station and may limit the interpretations that can be made from results using the blood sample analysis. Nevertheless, BMI and systolic and diastolic blood pressure did not differ significantly between participants attending and not attending blood sample drawing, with the exception that BMI was significantly different among men in Oslo.

## 5. Conclusions

In spite of the high BMI in Oslo, the study demonstrated that most CVD risk factors among women from Hargeisa were higher when compared to those among Somali women living in Oslo, with somewhat similar differences seen in men. However, the overall predicted CVD risk was not significantly different between the two locations. In addition, as overweight/obesity, smoking, hypertension, and high cholesterol were observed at concerning prevalence levels, intervention efforts are necessary and should be supported by continuing to monitor CVD risk factors in both populations.

## Figures and Tables

**Figure 1 ijerph-16-02353-f001:**
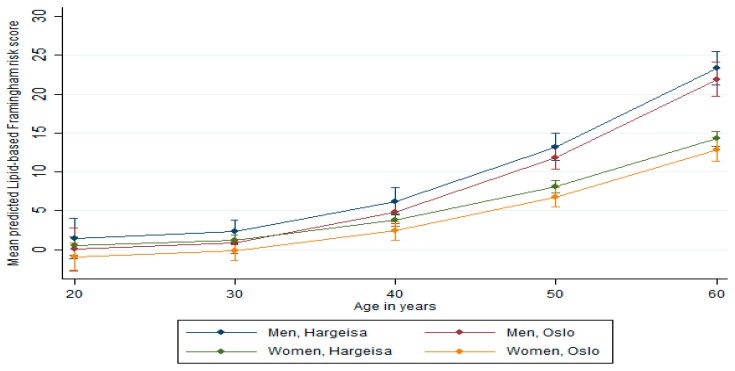
Lipid-based Framingham risk score by age, gender, and location (vertical lines are means with 95% confidence intervals).

**Figure 2 ijerph-16-02353-f002:**
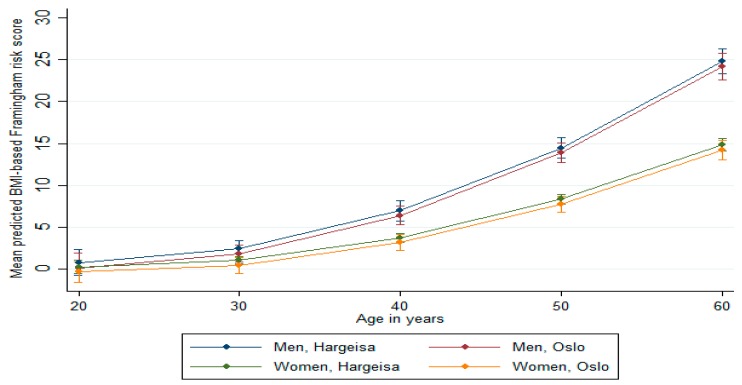
Body mass index (BMI)-based Framingham risk score by age, gender, and location (vertical lines are means with 95% confidence intervals).

**Figure 3 ijerph-16-02353-f003:**
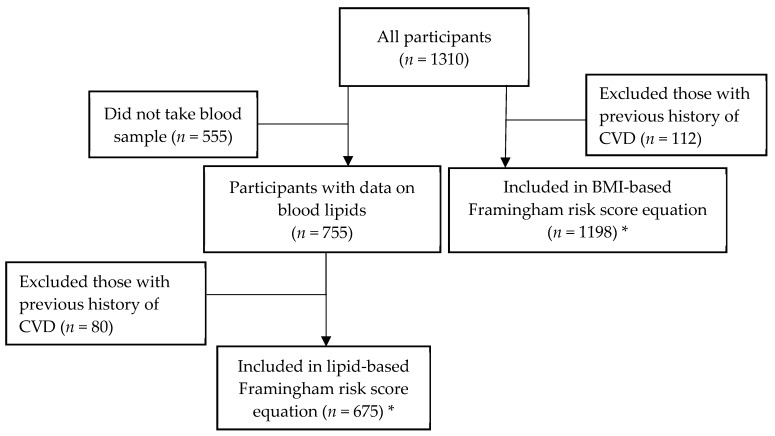
Flow chart illustrating participants included in the different analyses. * *n* is different between those who were included in the lipid-and BMI-based Framingham risk score equations due to missing data from lab results in the lipid-based analyses.

**Table 1 ijerph-16-02353-t001:** Characteristics of the study population in Oslo and Hargeisa (unadjusted).

**Men**	**Oslo (*n* = 106)**	**Hargeisa (*n* = 145)**	***p*** **Value**
Age (years), Mean ± SD	39.4 ± 11.0	38.2 ± 6.1	0.493
Education (%) *	89.4	52.4	<0.001
Employment (%) **	19.2	32.2	0.023
Marital status (%) ***	64.4	56.6	0.211
Body mass index, Mean ± SD	25.9 ± 3.4	22.1 ± 4.6	<0.001
SBP (mmHg), Mean ± SD	126.0 ± 17.0	129.7 ± 27.8	0.185
DBP (mmHg), Mean ± SD	81.3 ± 10.3	82.7 ± 12.7	0.349
**Women**	**Oslo (*n* = 106)**	**Hargeisa (*n* = 953)**	***p*** **Value**
Age (years), Mean ± SD	37.5 ± 9.6	39.1 ± 14.1	0.137
Education (%) *	48.6	14.3	<0.001
Employment (%) **	22.3	88.4	<0.001
Marital status (%) ***	62.9	78.4	<0.001
Body mass index, Mean ± SD	30.0 ± 6.7	27.2 ± 5.9	<0.001
SBP (mmHg), Mean ± SD	118.1 ± 19.8	128.2 ± 24.6	<0.001
DBP (mmHg), Mean ± SD	78.3 ± 9.7	83.8 ± 12.5	<0.001

* Education = % completed secondary school or more. ** Employment = % who are unemployed. *** Marital status = % currently married. SBP: systolic blood pressure; DBP: diastolic blood pressure.

**Table 2 ijerph-16-02353-t002:** Mean serum lipids (mmol/L) by location and gender adjusted for age ^a^.

Variables	Men			Women		
	Oslo (*n* = 76)	Hargeisa (*n* = 60)	Mean Difference ^b^ (95% CI)	Oslo (*n* = 85)	Hargeisa (*n* = 534)	Mean Difference ^b^ (95% CI)
Fasting serum glucose (mmol/L)	5.3	5.4	−0.1 (−0.6, 0.3)	5.2	5.4	−0.2 (−0.6, 0.3)
Triglycerides (mmol/L)	1.1	1.6	−0.5 (−0.8, −0.1)	0.9	1.3	−0.4 (−0.5, −0.2)
Total cholesterol (mmol/L)	4.6	4.5	0.1 (−0.2, 0.3)	4.5	4.7	−0.2 (−0.3, 0.02)
HDL cholesterol (mmol/L)	1.2	1.00	0.2 (0.1, 0.2)	1.4	1.1	0.3 (0.2, 0.3)
LDL cholesterol (mmol/L)	3.0	2.7	0.3 (0.1, 0.6)	2.9	2.8	0.1 (−0.1, 0.3)
TC/HDL ratio	3.9	4.4	−0.5 (−0.8, −0.2)	3.3	4.1	−0.8 (−1.0, −0.6)

**^a^** model was evaluated at age 40 years. ^b^ (Oslo minus Hargeisa). HDL: high-density lipoprotein; LDL: low-density lipoprotein; TC: total cholesterol.

**Table 3 ijerph-16-02353-t003:** Prevalence (%) of risk factors among men and women from Oslo and Hargeisa, age adjusted ^a^.

Risk Factors	Prevalence, % (95% CI)		
	Oslo	Hargeisa	*p* Value
**Men**			
Overweight/obese (BMI ≥ 25 kg/m^2^)	61.5 (52.0, 71.1)	22.9 (15.7, 30.0)	<0.001
Hypertension (SBP ≥ 140 mmHg or DBP ≥ 90 mmHg and/or medicated)	34.1 (24.6, 43.5)	32.9 (24.5, 41.3)	0.860
Diabetes (FSG ≥ 7.0 mmol/l and/or medicated)	5.4 (6.8, 10.2)	8.3 (4.3, 16.6)	0.467
High cholesterol (TC/HDL ratio ≥ 5 and/or medicated)	19.1 (10.2, 28.0)	35.0 (22.6, 47.4)	0.040
Current smokers	20.1 (12.4, 27.9)	27.1 (19.8, 34.4)	0.209
Alcohol consumption	8.6 (3.2, 14.1)	0	n/a
**Women**			
Overweight/obese (BMI ≥ 25 kg/m^2^)	78.1 (70.4, 85.9)	64.3 (61.1, 67.5)	0.005
Hypertension (SBP ≥ 140 mmHg or DBP ≥ 90 mmHg and/or medicated)	17.4 (9.9, 25.0)	34.9 (31.5, 38.4)	0.001
Diabetes (FSG ≥ 7.0 mmol/l and/or medicated)	9.4 (3.1, 15.7)	11.1 (8.3, 13.9)	0.649
High cholesterol (TC/HDL ratio ≥ 5 and/or medicated)	6.0 (1.0, 11.1)	21.0 (17.4, 24.6)	0.003
Current smokers	0	0	
Alcohol consumption	0	0	

**^a^** model was evaluated at age 40 years. FSG: fasting serum glucose.

**Table 4 ijerph-16-02353-t004:** Predicted 10-year cardiovascular disease (CVD) risk of ≥10% using the Framingham risk scores based on lipids and BMI, age adjusted.

Predicted Risk	Men			Women		
**Lipid-based**	**Oslo** **(*n* = 73)**	**Hargeisa** **(*n* = 50)**	***p*** **Value**	**Oslo** **(*n* = 82)**	**Hargeisa** **(*n* = 470)**	***p*** **Value**
Predicted 10-year risk ≥ 10%, % (95% CI), at age 45 years	22.8 (8.8, 36.8)	49.6 (11.5, 87.7)	0.118	8.5 (1.3, 15.6)	8.4 (4.5, 12.5)	0.987
**BMI-based**	**Oslo** **(*n* = 101)**	**Hargeisa** **(*n* = 134)**	***p*** **Value**	**Oslo** **(*n* = 103)**	**Hargeisa** **(*n* = 860)**	***p*** **Value**
Predicted 10-year risk ≥ 10%, % (95% CI), at age 45 years	41.4 (27.0, 55.7)	54.4 (33.1, 75.8)	0.335	9.1 (0.1, 16.3)	8.6 (5.9, 11.3)	0.895

**Table 5 ijerph-16-02353-t005:** Estimated Framingham risk score by education, age adjusted ^a^.

Level of Education	Lipid-Based Framingham Risk Score			BMI-Based Framingham Risk Score		
**Men**	***N*** **= 123**	**Mean (95% CI)**	***p*** **Value**	***N*** **= 235**	**Mean (95% CI)**	***p*** **Value**
Education						
< Secondary school	35	5.5 (3.0, 7.9)	0.890	73	5.8 (4.3, 7.4)	0.149
≥ Secondary school	88	5.3 (3.0, 7.4)		162	7.2 (5.6, 8.8)	
**Women**	***N*** **= 552**	**Mean (95% CI)**	***p*** **Value**	***N*** **= 963**	**Mean (95% CI)**	***p*** **Value**
Education						
< Secondary school	445	3.6 (3.0, 4.3)	0.078	781	3.7 (3.2, 4.2)	0.897
≥ Secondary school	107	3.1 (2.3, 3.9)		182	3.8 (3.0, 4.5)	

**^a^** model was evaluated at age 40 years. < Secondary school: did not complete secondary school; ≥ Secondary School: completed secondary school or more.

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
