# Peer review of "Comparison of Cardiovascular Risk Factors among Somalis Living in Norway and Somaliland"

_ijerph, 2019, doi:10.3390/ijerph16132353_

Reviewer 1 Report

           The authors conducted a study to compare CVD risk factors and future risk of CVD among Somalis living in Norway and Somaliland.  The paper is very well written, well organized, and interesting.  They used appropriate statistical methods and provided an adequate discussion of the results.  I have a few minor editorial comments that will hopefully improve the paper.

           Introduction, page 2, lines7-8:  This sentence can be rephrased as follows: "...despite the concentration of Somalis in Norway and suggestions of an increased CVD risk among ethnic minority groups, little research has been conducted on CVD risk factors in the country...". 

           Data collection, line 23:  Use 'readings' instead of 'reading'.

           Discussion, 3rd para., line 8:  Use 'higher' instead of 'high'.  Also, line 33:  Change beginning of sentence to "No relationship was found between the mean scores....".

Reviewer 2 Report

The authors present an interesting manuscript about the prevalence of cardiovascular (CVD) risk factors and predicted 10-year risk of CVD using Framingham risk score models among Somalis living in Norway and Somaliland. As authors claimed, there are limited data on cardiovascular risk factors in this ethnic group. One of the study limitations mentioned by the authors is representativeness of the selected samples; however, taking in mind the difficulties in organizing epidemiological surveys in such populations, the presented data could be used for approximate description of the situation.

I have only a few comments that need to be clarified to provide reasonable information for the readers.

           1. How alcohol consumption was assessed? Does it mean that the respondents never use any type of alcoholic drink?

          2. The authors need to check the number of respondents in the tables and text. There are some discrepancies. For example, it is written in the Results section, I paragraph: ‘A total of 1310 participants were included in the analysis. Women from Hargeisa composed the largest sub-group with 953 participants, while 106 women participated from Oslo. Among men, there were 145 participants from Hargeisa and 106 from Oslo.’ The last paragraph of the Methods section states: ‘For the lipid-based equation, we included a total of 123 men (73 from Oslo and 50 from Hargeisa) and 552 women (82 from Oslo and 470 from Hargeisa). In the BMI-based equation, a total of 235 men (101 from Oslo and 134 from Hargeisa) and 963 women (103 from Oslo and 860 from Hargeisa) were included.’

Reviewer 3 Report

             Excellent study. There are only few available data extensively and properly comparing African natives leaving in their country with those who migrated to Europe. 

            The authors should probably add a few words about the rather different family incomes across both settings. They already did it partly while comparing oils from both locations,In turn, the authors should add their interpretation of similar CV risk factors in spite of very different lifestyles, nutritional habits and educational levels.
